


# Compound flood events: different pathways–different impacts–different coping options?

Annegret H. Thieken[1], Guilherme S. Mohor[1], Heidi Kreibich[2], Meike Müller[3]

[1]Institute of Environmental Science and Geography, University of Potsdam, Potsdam-Golm, 14476, Germany
[2]GFZ German Research Centre for Geosciences, Section Hydrology, Potsdam, 14473, Germany
[3]Deutsche Rückversicherung AG, Düsseldorf, 40549, Germany

*Correspondence to*: Annegret H. Thieken (annegret.thieken@uni-potsdam.de)

**Abstract.** Several severe flood events hit Germany in recent years, with events in 2013 and 2016 being the most destructive ones although dynamics and flood processes were very different. While the 2013-event was a slowly rising widespread fluvial flood accompanied by some severe dike breaches, the events in 2016 were fast onset pluvial floods, which resulted in some places in surface water flooding due to limited capacities of the drainage systems and in others, particularly in small steep catchments, in destructive flash floods with high sediment loads. Hence, different pathways, i.e. different routes that the water takes to reach (and potentially damage) receptors, in our case private households, can be identified in both events. They can thus be regarded as spatially compound flood events. This paper analyses how affected residents coped with these different flood types (fluvial and pluvial) and their impacts while accounting for the different pathways (river flood, dike breach, surface water flooding and flash flood) within the events. The analyses are based on two data sets with 1652 (for the 2013-flood) and 601 (for the 2016-flood) affected residents who were surveyed around nine months after each flood, revealing little socio-economic differences–except for income–between the two samples. The four pathways showed significant differences with regard to their hydraulic and financial impacts, recovery, warning processes as well as coping and adaptive behaviour. There are no or just small differences with regard to perceived self-efficacy and responsibility offering entry points for tailored risk communication and support.

## 1 Introduction

Floods are the most frequent natural hazard worldwide affecting the most people (CRED and UN-DRR, 2020), with Europe being no exception (EEA, 2019). Particularly, river floods have caused substantial losses in many European countries in recent years calling for better risk reduction strategies. Among others, severe flooding in 2002 caused losses of over EUR 21 billion in Central Europe (EEA, 2019) and triggered the development of the Floods Directive (2007/60/EC), a European framework for reducing flood impacts by integrated risk management approaches. In Germany, the Floods Directive has been implemented for river and coastal floods. Together with further lessons learned from past flood events (e.g., DKKV, 2003; 2015), this has already led to substantial improvements in the management of river floods. For example, several high-frequency-low-impact-events, e.g. in 2005, 2006, 2010 and 2011, revealed that regional and local governments in Germany,



as well as flood-prone residents and companies, have been adapting to flood risk with enhanced precaution and preparedness (Kreibich and Thieken, 2009; Kreibich et al., 2011; Kienzler et al., 2015). In addition, the widespread flood of June 2013 demonstrated improved flood risk management all over Germany by the fact that this most severe flood event in hydrological terms (Merz et al., 2014; Schröter et al., 2015) caused lower losses, i.e. EUR 6 to 8 billion, than the 2002-flood

with EUR 11.6 billion (Thieken et al., 2016a,b). Still, some areas, particularly those affected by dike breaches, suffered from severe losses. In general, buildings affected by dike breaches tend to experience higher losses than buildings affected by usual fluvial flooding (Cammerer and Thieken, 2011; Mohor et al., 2020). Following the source-pathway-receptor-consequences model (SPRC-model; e.g. Sayers et al., 2002), dike breaches can be regarded as a specific pathway within a regional flood event, since the floodwater takes a different route to reach (and potentially damage) receptors such as

buildings or residents. Except for Vogel et al. (2018) and Mohor et al. (2020), such pathways have been rarely studied in impact analyses or loss modelling although there are indications that the resulting consequences differ.

Pluvial flooding has occurred in several places in Germany in recent years, e.g. in the city of Münster in 2014 (Spekkers et al., 2017) or in the village of Braunsbach in 2016 (Bronstert et al., 2018), causing damage that was unprecedented for this type of flooding. Particularly the event of May/June 2016 challenged water authorities and residents: several places in

Germany were affected by heavy rainfall and hail leading to surface water flooding due to limited capacities of urban drainage systems (GDV, 2016; Piper et al., 2016). Moreover, in some places, particularly in the small towns of Braunsbach and Simbach, flooding was accompanied by quick concentrated surface runoff activating huge amounts of mud, debris and further material that was carried downstream and threatened people and assets (Piper et al., 2016; Laudan et al., 2017; Vogel et al., 2017). Overall losses amounted to EUR 2.6 billion (Munich Re, 2017), eleven people lost their lives and more than 80

people were injured, mostly by lightning strokes.

The events of 2016 also impacted policy processes on pluvial flood risk management (e.g., Kind et al., 2019; Riese et al., 2019; Thieken et al., 2019). Analyses of pluvial floods illustrate that warning is more difficult and residents tend to be less experienced with this flood type and are hence less prepared for it, but average property losses are commonly lower in comparison to fluvial floods (see Kienzler et al., 2015; Rözer et al., 2016; Spekkers et al., 2017; Kind et al., 2019; GDV,

2020). These analyses, however, mainly focus on surface water flooding in urban areas, ignoring that impacts caused by flash floods can be exceptionally high (GDV, 2016; Laudan et al., 2017). The severity of flash flood processes also affect mental health as well as precautionary behaviour (Laudan et al., 2020) and have even lead to relocations of some buildings at risk, a risk management strategy that has been rarely implemented in Germany (Mayr et al., 2020). Hence, to better understand flood impacts and coping options, it seems necessary to not only distinguish different flood types (fluvial and

pluvial flood), but also different pathways within one flood event, like dike breaches and flash floods. Following Zscheischler et al. (2020) we refer to the flood events in 2013 and 2016 as spatially compound flood events (see also section 2), since they represent a situation in which multiple locations are impacted within a limited time window and are connected via a physical modulator, i.e. the atmospheric circulation. Zscheischler et al. (2020) further recommend separating and analysing different elements, i.e pathways in our view, to better understand the event as a whole. Therefore, we created


subsamples that capture different flood pathways, i.e. dike breaches, river floods, flash floods and surface water flooding, to study their characteristics within and between the two flood events of 2013 and 2016 (see section 3.3 and Figure 1). We hypothesize that such in-depth analyses of impact and coping patterns of different flood types and pathways provide entry points to better tailor flood risk management to local circumstances.

Analyses of data from flood events between 2002 and 2013 suggest that flood pathways play an important role when it
comes to the assessment of (financial) flood impacts (Vogel et al., 2018; Mohor et al., 2020). Differences in coping options during the event as well as in recovery in its aftermath are less clear. Therefore, this paper aims to reveal whether and how people affected by different flood types and pathways were prepared before the damaging event, how they were impacted in hydraulic, financial and psychological terms, and how they coped with and recovered from these impacts. The intention is to provide more insights that help establish risk management strategies tailored to different flood types and pathways. Like in
previous studies (Thieken et al., 2007; Kienzler et al., 2015, for fluvial floods and Rözer et al., 2016; Spekkers et al., 2017, for pluvial floods) the risk management cycle is used as guiding framework. However, in contrast to the previous studies this paper also looks at patterns within the flood events separating cases affected by dike breaches in 2013 and flash floods with heavy sediment load in 2016 from the overall samples to better understand the impacts of and coping options towards specific flood pathways. For clarity, general flood types are termed fluvial and pluvial floods in this paper, while pathways
within the events are named dike breach, river flood, surface water flood and flash flood.

**2 Compound Flood Events**

Accounting for interactions between hazard processes helps to better understand and prepare for complex events. Commonly, compound, interacting and cascading events are distinguished (e.g., Pescaroli and Alexander, 2018). Originating from research on climate change, compound events are described as (1) simultaneous or successively occurring (climate-
related) events such as simultaneous coastal and fluvial floods; 2) events combined with background conditions that augment their impacts such as rainfall on already saturated soils; or (3) a combination of (several) average values of climatic variables that result in an extreme event (IPCC, 2012; Pescaroli and Alexander, 2018). Besides the coincidence of coastal and fluvial flooding which is commonly referred to as compound flood event, there are more combinations that can be termed like that. For example, in August 2002, the city of Dresden in Saxony, Germany, was hit by four consecutive flood waves, which were
all triggered by the same rainfall event: first, surface water flooding occurred in the city as an immediate response to the heavy precipitation on 12 August 2002 and the limited capacity of the sewer system, which was shortly, i.e. one day later, followed by a flash flood from the local and mid-sized rivers Weißeritz and Lockwitzbach that drain into the bigger river Elbe within the city area of Dresden. A few days later, i.e. on 17 August 2002, this flooding was followed by inundations from the flood wave of the river Elbe, which was later followed by high groundwater levels lasting for several months
(Kreibich et al., 2005). Further examples for compound flood events are similar rainfall amounts that can lead to different flood situations depending on the antecedent soil moisture and the characteristics of the catchment (e.g., topography, size,



land use). Zscheischler et al. (2020) termed such situations spatially compound events, which we assigned to the flood events of 2013 and 2016.

## 2.1 The flood in June 2013

In June 2013, widespread fluvial flooding occurred in Central Europe, particularly in Germany: twelve out of 16 German federal states were affected; eight of them declared a state of emergency (BMI, 2013 as cited in Thieken et al., 2016a,b). Flooding was triggered by a combination of wet antecedent conditions and high precipitation amounts between 31 May and 2 June 2013 (Merz et al., 2014; Schröter et al., 2015). By the end of May 2013, record-breaking antecedent soil moisture was recorded in 40 % of the German territory (DWD, 2013) and above-average initial streamflows were observed in many rivers

(Thieken et al., 2016a). Hotspots of precipitation between 31 May and 3 June 2013 totalled up to 346 mm within 72 hours at the DWD weather station of Aschau-Stein (Schröter et al., 2015). This combination resulted in high flood peaks in the upper catchments of the rivers Rhine and Weser and particularly in many parts of the catchments of the rivers Danube and Elbe (Thieken et al., 2016a). Altogether, peak flows exceeded the five-year flood discharge in 45 % of the German river network (Schröter et al., 2015). Around 1,400 km of the river network saw 100-year flood discharges. Hydrological and statistical

analyses indicated that this event was Germany's most severe flood over the past 60 years (Merz et al., 2014) leading to widespread inundations, particularly along the rivers Danube and Elbe. Although huge investments had been made in upgrading embankments after the 2002-flood, some dike breaches and consequent inundations of their hinterland occurred. Three breaches were particularly severe (Merz et al., 2014): (1) a breach at Deggendorf-Fischerdorf at the confluence of the rivers Isar and Danube flooded several properties; due to floating and bursting oil tanks and consequently highly

contaminated flood water, 150 homes had to be completely rebuilt (Bavarian Parliament, 2014); (2) a breach in Klein Rosenburg-Breitenhagen at the confluence of the rivers Saale and Elbe and (3) a breach near Fischbeck at the middle reach of the Elbe River that also affected the high-speed train connection between Berlin and Hanover which was disrupted for several months (Thieken et al., 2016b). In all of Germany, 14 people died and direct losses summed up to EUR 6 to 8 billion (Thieken et al., 2016b).

In comparison to regions flooded by a river, areas affected by dike breaches tend to suffer from extended inundation durations (Vogel et al., 2018) and – where oil heating is common – floating and leaking oil tanks that cause considerable material and environmental damage (DKKV, 2015; Thieken et al., 2016b). Considering the triggering mechanism of this flood, as well as the dike breaches mentioned above, this event can be understood as a spatially compound event. To account for different flood pathways, residents affected by "normal" river floods and residents affected by dike breaches are analysed

separately in this paper.

## 2.2 Flooding in May and June 2016

From May 26 to June 9, 2016, Germany and parts of central and southern Europe were hit by an extraordinarily high number of severe convective storms with intense rainfall and hail. This thunderstorm episode was caused by the interaction of high





atmospheric moisture content, low thermal stability, weak wind speed and large-scale lifting by surface lows (Piper et al.,
2016). Low wind speed at mid-tropospheric levels led to nearly stationary or slow-moving convective cells and hence to
locally extreme rain accumulations exceeding 100 mm within 24 hours. Due to atmospheric blocking these boundary
conditions persisted over almost two weeks (Piper et al., 2016). Depending on the characteristics of the affected catchments
and areas the heavy precipitation triggered surface water flooding (due to limited sewer capacity, e.g. in the city of Hanover),
inundations along small rivers and creeks and flash floods, partly carrying huge amounts of mud and debris. The main
hotspots occurred in South Germany. In Braunsbach a small village in Baden-Wuerttemberg the extreme precipitation of
more than 100 mm within 2 hours on May 29 caused a devastating flash flood (Bronstert et al., 2017). The Orlacher Bach, a
creek that runs through the village with just 6 km² catchment size and very steep slopes, showed extreme runoff with
massive debris transport of 42,000 m³ (Vogel et al., 2017). Streets were blocked with gravel and stones up to a thickness of 2
to 3 m producing immense damage to buildings and infrastructure (Laudan et al., 2017). In Simbach a village in south
Bavaria situated on the river Inn the rainfall amounted to 120 mm in 24 hours on June 1 (Pieper et al, 2016). Subsequently
the small river Simbach (33 km² catchment size) and its tributaries showed extreme runoff. At the gauging station Simbach
the water level rose from 50 cm to 506 cm within 14 hours. Several culverts were blocked with debris and driftwood, dams
broke and parts of the village were flooded (LfU, 2017).

In all of Germany 11 people died and the economic loss amounted to EUR 2.6 billion which is extraordinary high with
regard to heavy rainfall and thunderstorms in Germany (GDV, 2016; Laudan et al., 2017; Munich Re, 2017; Vogel et al.,
2017). Because of the huge losses in Simbach and other villages in Bavaria a grant and loan programme for compensating
flood damage to residential buildings and household contents was implemented (Bavarian State Government, 2016). In
Baden-Wurrtemberg, the market penetration of insurance against natural hazards is still high, i.e. around 94%, due to the fact
that it was mandatory until 1994 (Surminski and Thieken, 2017; GDV, 2020).

Since different types of flooding and various runoff dynamics could be observed from May 26 to June 9, this event is also
treated as a spatially compound flood in this paper. The dynamics comprise different pathways, flow velocities, water depths
as well as different impacts that are difficult to categorise distinctly. Yet, households have been mainly affected by shallow
surface water flooding, but, in fewer cases, also by the forceful overflowing of water bodies and partly dike breaches which
led to strong flash floods with a heavy sediment load (e.g. Braunsbach and Simbach). Thus, the data set from this pluvial
flood was separated into cases affected by low/medium surface water flooding on the one hand and cases that suffered from
flash floods on the other hand (Figure 1; see section 3.3 for details).

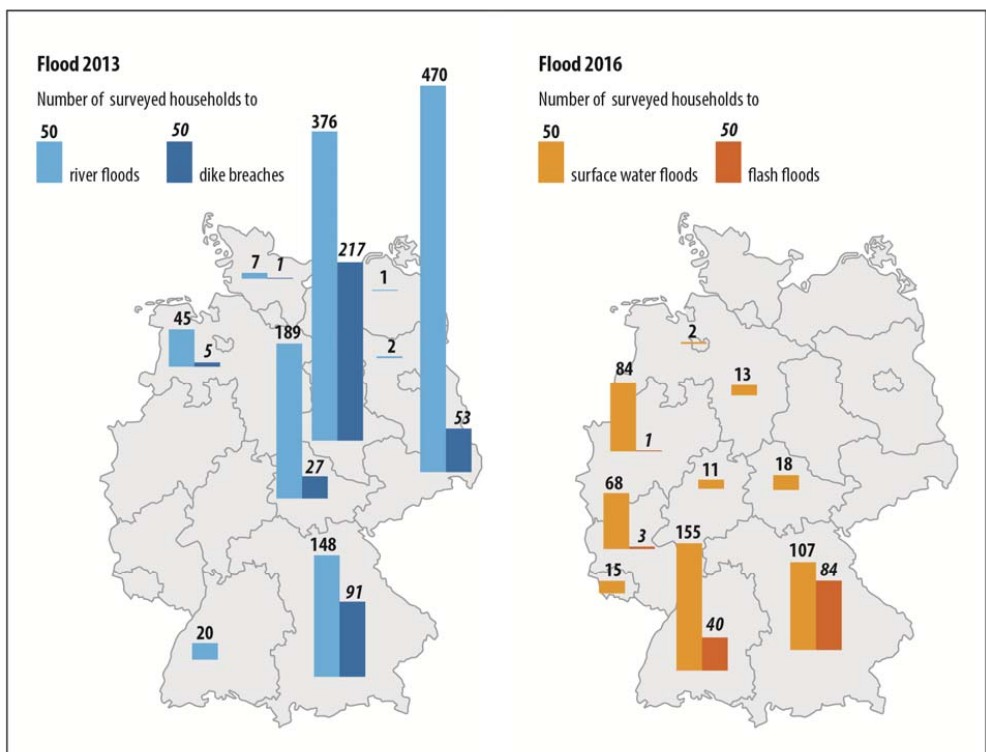

**Figure 1: Geographic overview of the number of households surveyed about the flood of 2013 (left) and 2016 (right).**

### 3 Data and Methods

The analyses are based on survey data that were gathered among private households that suffered from property damage caused by flooding in 2013 or 2016. Both surveys were conducted around nine months after the respective damaging event using computer-aided telephone interviews (CATI), during which residents were guided through a standardized questionnaire (see Thieken et al., 2017). On average, an interview lasted around 30 minutes.

### 3.1 Sampling flood-affected households

To identify affected households, media reports and satellite images were used to compile a list of inundated streets and zip codes. In some cases, this information was provided by affected communities and districts or fire brigades. The lists served as a basis for retrieving telephone numbers (landlines) from public telephone directories. Due to a high number of non-





affected residents within the inundated areas, all retrieved telephone numbers were finally called in order to create comprehensive samples. Always the person in the household who had the best knowledge about the flood event was questioned. The surveys were conducted by a subcontracted pollster from 18 February to 24 March 2014 for the 2013-flood and from 28 March to 28 April 2017 for the 2016-event. In total, 1652 interviews were completed for the 2013-flood (out of a total of 43,281 numbers, from which 16,554 could not be reached during the field time; another 16,721 residents did not suffer from financial damage and 8144 refused to participate). For the 2016-event, it was possible to complete 601 interviews (out of 42,487 retrieved numbers, from which 24,486 could not be reached during the field time; 12,010 residents did not suffer from financial damage and 4254 refused to participate).

### 3.2 Contents of the questionnaire and data processing

The questionnaires already presented by Thieken et al. (2005; 2007) and Rözer et al. (2016) were slightly adapted for the two surveys. The questionnaires contained about 160 questions addressing a range of topics: depth, velocity and duration of the inundation at the affected property, contamination of the flood water, flood warning, emergency measures, characteristics of and amount of damage to household contents and buildings, recovery and psychological burden of the interviewed person, precautionary measures, previously experienced flood events, perceived threat and coping appraisal, as well as socio-demographic information. In addition, the 2013-questionnaire addressed evacuation and cleaning-up and asked for an assessment of the (governmental) disaster aid (Thieken et al., 2016b); these items were not included in the questionnaire about the 2016-event. In both surveys, tenants were only asked about their household, the damage to contents and some core characteristics of the building. Several questions used a Likert-type scale from 1 to 6, where "1" described the best option and "6" the worst. These options were explicitly verbalized; intermediate ranks could be used to graduate the assessment.

After the collection, data was post-processed through comparison and consistency checks. Some of these were already performed during the interview, e.g., concerning questions about some characteristics of the building and the type of the losses. Additional checks were performed in the aftermath, e.g., the size of the household was compared to the reported numbers of children and elderly in that household. In addition, some items were aggregated to indicators as described by Thieken et al. (2005) and Laudan et al. (2020): contamination, source of the flood warning, emergency measures (short-term; performed during the event), precautionary measures (long-term measures, implemented before or after the flood) and previously experienced flooding.

Further, the total asset values of contents and buildings were estimated based on the floor space (of the building or the flat) and standardized values as proposed in guidelines of the German insurance industry. For contents, a standard value of 650 EUR/m² as of 2005 was scaled to the year of the event by a consumer price index excluding food, resulting in 695.90 EUR/m² (as of 2013) and 719.52 EUR/m² (as of 2016). The total value of a building was estimated by the "Mark1914"-insurance value per m² per building type multiplied by the "Gleitender Neuwertfaktor" (16.2 for 2013 and 17.2 for 2016), a specific building price index used by the German insurance industry. If the reported damage exceeded the so-estimated asset value, a loss ratio of 1 was assumed, i.e. the asset value was substituted by the reported financial loss.



### 3.3 Subsamples

To study differences in flood pathways the following subsamples were distinguished (compare Figure 1):

- 2013-dike breaches: all households that reported that they had been affected by a dike breach were included in this subsample; this applied to 394 cases, i.e. to around 24 % of all surveyed cases affected by flooding in 2013;

- 2013-river flooding: all other households from the 2013-data set, i.e. 1258 cases (76 %);

- 2016-flash floods: all surveyed households from areas that had been severely affected by flash floods and sediment load; this applied to 128 cases, i.e. to around 21 % of all surveyed cases affected by flooding in 2016;

- 2016-surface water flooding: all other households from the 2016-data set, i.e. 473 cases (79 %).

Flash floods in 2016 were identified by means of quantitative and qualitative methods (Laudan et al., 2020). First, hourly
data from rain gauges was obtained from the Climate Data Center of the German Meteorological Service (Deutscher Wetterdienst, DWD; https://cdc.dwd.de/portal/) for May and June 2016, considering all affected municipalities. If the rainfall within a district exceeded 25 mm per hour, it was marked to be potentially affected by a flash flood since according to the DWD definitions, local rainfall of more than 25 mm per hour initializes a severe weather warning. Second, on the basis of online literature, photos, local press articles, and media attention, a general flood intensity classification (low,
medium, strong) was created for each affected city. In a final step, based on the rainfall information and the review of online sources, surveyed households were classified as surface water flooding if comparatively low flood impacts were documented in the municipality and to flash floods if high flood impacts were mentioned. The resulting classification was cross-checked by survey answers to a question on flood pathways revealing that cases that were classified as flash floods were more often accompanied by overflowing water bodies, forceful surface runoff and dike breaches, while cases that were classified as
surface water floods dominantly reported overflowing sewer systems, surface runoff and rising groundwater as source of flooding.

### 3.4 Data Analysis

Data subsets were compared either through the nonparametric Mann-Whitney-Wilcoxon two-sample test or Chi-Squared contingency table test, depending on whether a variable was metric or categorical (Noether, 1991), comparing the median of
225 differences or the closeness of expected frequencies, respectively.

A p-value threshold was set to 0.05 for statistical significance, regardless of the absolute difference or effect size. These procedures were run with R language (R Core Team, 2017) – with the assistance of the packages "stats", "rcompanion", and "PMCMR". For those variables, for which significant differences were revealed, further frequency analyses and descriptive statistics were calculated in SPSS. Means and frequencies are presented in relation to valid answers, i.e. ignoring no answers
or "I don't know" entries.



## 4 Results and Discussion

In this section we present the main differences and commonalities between and within the two flood events. Per topic we will first compare the fluvial 2013-flood to the pluvial 2016-flood, which is then followed by a comparison of the flood pathways within each event, i.e. river floods versus dike breaches for the 2013-event as well as surface water floods versus flash floods in 2016.

### 4.1 Socio-demographic characteristics of the subsamples

This section presents the characteristics of the surveyed residents in the four subsamples and introduces the approach of how we analyse differences between and within the two flood events. Besides the mean values for each item and each subsample as well as for the whole data set, Table 1 provides the test statistics of the Mann-Whitney-Wilcox or Chi-Square tests when comparing all data from the 2013-flood with the 2016-flood as well as when comparing the two subsamples (pathways) within each event.

On a 5%-significance level, Table 1 reveals that socio-demographic characteristics do not differ between the two events, except for the share of households with a monthly net income below EUR 1500 and the share of one-family homes. Both values are higher for the 2013-flood, reflecting that more rural areas were affected by this widespread fluvial flood. Those affected by surface water flooding in 2016 have the smallest percentage of households with income below EUR 1500 or, in other words, a higher share of higher-income households than the other subsamples, as well as the smallest percentage of households living in one-family homes reflecting that mainly urban areas were affected by this flood pathway. In contrast, the flood of 2013 widely affected rural areas in the Eastern parts of the country. The 2013-sample contains many cases from Saxony and Saxony-Anhalt (see Fig. 1); in East Germany the mean monthly net income per household amounted to 2521 Euro in 2013, while it was 3297 Euro in West Germany that was hit by the 2016-floodings (Destatis, 2018; see Fig. 1). So, the differences in income in our data reflect the regional income pattern in Germany.

In addition, there are slight (i.e. low-significant) differences between the two events with respect to the mean household size and homeownership (Table 1). However, these variables differ more pronounced between the two subsamples of the 2013-flood: surveyed households affected by riverine flooding had the smallest household size and the lowest percentage of home/apartment ownership (80%), whilst those affected by dike breaches showed the highest percentage of homeowners (92%). Similarly, the 2013-river subsample shows a lower share of one-family homes type (51%) than the 2013-dike subsample (71%). This indicates that areas affected by dike breaches were mostly rural areas with owner-occupied dwellings and larger families, while other areas affected in 2013 are probably located in more urban settings, also showing a better education and a higher mean age. Similar, but statistically weaker differences were found for the 2016-event. Here the regions affected by flash floods tend to contain more one-family homes, a lower age and less people with a high-school graduation than areas affected by surface water flooding. Still, there are no significant differences in the living area per person among the subsamples, despite a range between 55 m² (river floods) and 65 m² (flash floods). Often, the flash flood





subsample did not show high statistical differences to other subsamples, even when presenting the highest or smallest means due to its small number of cases (Table 1).


**Table 1: Socio-demographic characteristics of households affected by different flood pathways in 2013 and 2016.**

| Subsample (Pathway) | River2013 | ↔ | Dike2013 | 2013 ↔ 2016 | Surface2016 | ↔ | Flash2016 | Overall |
|---|---|---|---|---|---|---|---|---|
| Sample size | 1258 | — | 394 | — | 473 | — | 128 | 2253 |
| Socio-economic and demographic variables | | | | | | | | |
| Female interviewee [%] | 59.1 | | 55.3 | | 57.7 | | 53.9 | 57.8 |
| Mean age of the interviewees [years] | 60.4 | **** | 57.1 | | 59.3 | * | 56.2 | 59.3 |
| People with high school graduation (*Abitur*) [%] | 34.9 | ** | 27.4 | | 35.4 | * | 24.6 | 33.1 |
| Mean household size [number of people] | 2.4 | **** | 2.7 | . | 2.6 | | 2.7 | 2.52 |
| Households with a monthly net income <1500 Euro [%] | 35.6 | | 35.4 | **** | 16.4 | | 19.6 | 30.3 |
| Mean living area per person [m²] | 55.1 | | 58.0 | | 60.3 | | 65.5 | 57.5 |
| Homeowners (house or apartments) [%] | 79.7 | **** | 92.1 | . | 84.8 | | 90.6 | 83.6 |
| One-family homes [%] | 50.9 | **** | 71.0 | **** | 38.8 | ** | 53.3 | 52.1 |

*Percentage or means only regarding valid values, i.e. answered entries.*

*Comparison of subsets or 2013 to 2016 in the middle columns, with P-value ranges from Mann-Whitney-Wilcox or Chi-Square tests represented as:* `Legend: '****' ≤0.001 '***' ≤0.005 '**' ≤0.01 '*' ≤0.05 '.' ≤0.1 ' ' ≤1`


Altogether, the characteristics of the four subsamples lie within previous studies' averages, though with varied sample sizes and from different regions in Germany, which should be taken into account when interpreting further results. Previous works compared the socio-economic characteristics of survey respondents to a city or a national census. Some differences are noticeable such as older households and a greater share of ownership among respondents, possibly because only fixed

landlines were consulted. Given the similarity of sampling methods, we expect similar biases in all sub-samples. For a more detailed discussion of potential biases, see the works of Kienzler et al. (2015), Rözer et al. (2016) and Spekkers et al. (2017).

**4.2 Flood characteristics**

The hydraulic impacts of the flood events on the affected buildings are presented in Table 2 in terms of water level, flood duration, flow velocity, and the presence of contamination by oil, which all differ significantly between the events of 2013

and 2016 as well as within the two events (except for flow velocity in the case of the 2013-flood). There is a clear difference in water level from surface water floods, which mostly affected only the cellar of houses (indicated by negative values in Table 2), followed by river floods to cases of dike breaches and flash floods, which showed the highest mean water levels.



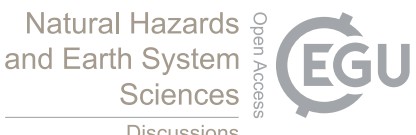
Negative average water levels, i.e. a water level below the ground surface, were also reported for pluvial and fluvial floods in 2005 (Rözer et al., 2016; Kienzler et al., 2015), a low river flood in 2011 (Kienzler et al., 2015) and the Danube area affected

in 2002 (Thieken et al., 2007). Hence, the mean water level roughly reflects the intensity of the event.

Surface water and flash floods have considerably shorter durations than river floods and dike breaches (Table 2). This pattern is also noticed by Kienzler et al. (2015), given that floods in 2002, 2006 and 2011 with an average duration of more than four days had a predominance of riverine flood dynamics, whilst Rözer et al. (2016) found shorter durations, less than one day in average, for pluvial floods. This pattern of the pathways is reflected in our samples.

Of those who were affected by river floods or dike breaches only around 15% reported a very high water velocity, i.e. a value of 5 or 6 on a scale from 1 to 6, in contrast to 63% in case of flash floods and 31% in case of surface water floods. The percentage of cases that reported oil contamination was the lowest in surface water floods (4%), followed by river floods (12%) and the flash flood subsample (24%). The highest value (34%) was reported by residents who were affected by dike breaches (see Table 2). A similar pattern is revealed for other contaminants like sewage, chemicals or petrol (Figure 2).

Table 2 illustrates that the people affected by different flood pathways had to cope with significantly different hazard situations, particularly in terms of water levels, flood duration and oil contamination. In addition, residents affected in 2016 by flash floods had to cope with high flow velocities. These findings confirm that our subsamples represent significantly differing flood pathways, while their socio-demographic characteristics differ comparatively little (see section 4.1). The next section looks into the financial flood impacts and recovery before we address coping options and strategies.


**Table 2: Hydraulic flood characteristics and impacts reported by households affected by different flood pathways in 2013 and 2016.**

| Subsample (Pathway) | River2013 | ↔ | Dike2013 | 2013 ↔ 2016 | Surface2016 | ↔ | Flash2016 | Overall |
|---|---|---|---|---|---|---|---|---|
| Sample size | 1258 | — | 394 | — | 473 | — | 128 | 2253 |
| Mean water level above top ground surface [cm] | 46.4 | *** | 76.3 | **** | -104 | **** | 86.3 | 23.7 |
| Mean flood duration [hours] | 173 | **** | 312 | **** | 41 | **** | 36 | 164 |
| Cases [%] that reported very high flow velocity, i.e. 5 or 6 on a scale from 1= no flow to 6 = very high velocity/ turbulent flow | 15.2 | | 15.5 | **** | 31.1 | **** | 63.1 | 21.1 |
| Cases that reported oil contamination [%] | 12.2 | **** | 34.3 | **** | 3.8 | **** | 24.2 | 15.0 |

*Percentage or means only regarding valid values, i.e. answered entries.*

*Comparison of subsets or 2013 to 2016 in the middle columns, with P-value ranges from Mann-Whitney-Wilcox or Chi-*
*Square tests represented as:* Legend: '*****' ≤0.001 '***' ≤0.005 '**' ≤0.01 '*' ≤0.05 '.' ≤0.1 ' ' ≤1

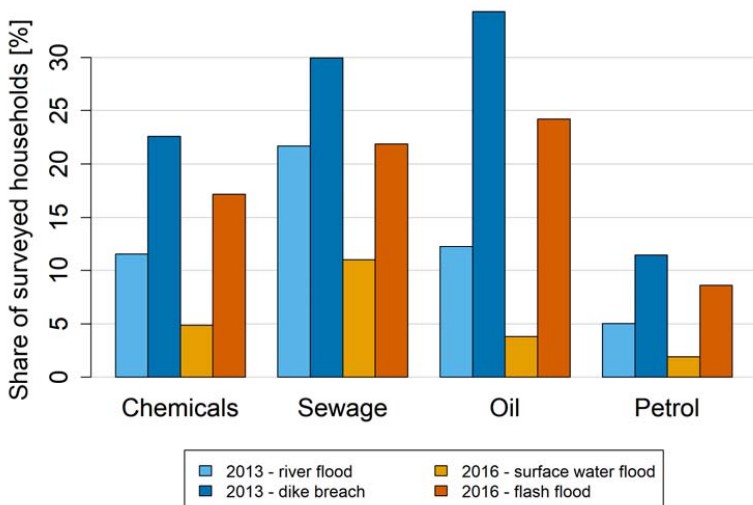

**Figure 2: Contaminants in the flood water as reported by households affected by different flood pathways in 2013 and 2016 (multiple answers possible).**

### 4.3 Financial flood impacts and perceived recovery

The average financial losses of buildings and of household contents differ significantly between and within the flood events (Table 3). Here, the financial loss refers to the repair and replacement costs (in prices of 2016). Residents affected by flash floods suffered from the highest financial losses – in absolute numbers as well as in terms of loss ratios, followed by those affected by dike breaches and river floods. Losses caused by surface water flooding resulted in the lowest amounts (in absolute numbers as well as with regard to loss ratios; see Table 3). Overall, the significant differences in the flood processes and the resulting hydraulic loads presented in Table 2 are reflected in the adverse effects of the floods.

To capture the status of recovery at the time of the survey, i.e. 8 to 10 months after the damage occurred, payments received to compensate losses were recorded. Further, respondents were asked to assess the accomplishment of the replacement of damaged household items or of the repair works at the damaged building on a Likert-scale. On a similar scale, they were asked to assess the psychological burden the flood still had at the time of the survey. Table 3 reveals that all variables except for the perceived status of the replacement of damaged household items significantly differ between 2013 and 2016. In addition, there are highly significant differences between the pathways within the two events. In general, respondents affected in 2016 received higher pay-outs, assessed their recovery a bit better and felt less burdened than those affected in 2013. However, those who experienced a flash flood in 2016 recovered less and felt more burdened than those affected by surface water flooding. Similarly, residents affected by dike breaches in 2013 are worse off than those affected by river flood.




**Table 3: Financial flood impacts, perceived recovery and psychological burden reported by households affected by different flood pathways in 2013 and 2016.**

| Subsample (Pathway) | River2013 | ↔ | Dike2013 | 2013 ↔ 2016 | Surface2016 | ↔ | Flash2016 | Overall |
|---|---|---|---|---|---|---|---|---|
| Sample size | 1258 | — | 394 | — | 473 | — | 128 | 2253 |
| **Financial damage** | | | | | | | | |
| Mean financial damage to the building [EUR] (1) | 48,610 | **** | 81,910 | **** | 19,720 | **** | 134,600 | 53,610 |
| Mean financial damage to the contents [EUR] (1) | 16,220 | **** | 27,830 | *** | 13,940 | **** | 51,080 | 20,200 |
| Mean loss ratio of the building [%] | 9 | **** | 17 | **** | 4 | **** | 23 | 11 |
| Mean loss ratio of the contents [%] | 19 | **** | 30 | **** | 12 | **** | 42 | 22 |
| **Perceived recovery AT THE TIME OF THE INTERVIEW, i.e. 8 to 10 months after the damaging flood event** | | | | | | | | |
| Mean loss compensation (payouts) [EUR] (1) | 10,810 | **** | 18,200 | **** | 16,770 | **** | 36,260 | 13,670 |
| Mean perceived status of repair works at the building [Likert-scale from 1 (building is completely restored) and 6 (there is still considerable damage)] | 2.8 | **** | 3.3 | **** | 1.7 | **** | 2.9 | 2.6 |
| Mean perceived replacement of damaged household items [Likert-scale from 1 (damaged household items are completely replaced) and 6 (still considerable missing household items)] | 2.4 | **** | 3.0 | . | 2.2 | **** | 3.2 | 2.6 |
| Mean perceived psychological burden [Likert-scale from 1 (no burden at all) to 6 (still heavy burden)] | 3.4 | **** | 4.0 | **** | 2.6 | **** | 3.7 | 3.3 |

*Percentage or means only regarding valid values, i.e. answered entries.*

*(1) In 2016-prices*

*Comparison of subsets or 2013 to 2016 in the middle columns, with P-value ranges from Mann-Whitney-Wilcox or Chi-Square tests represented as:* `Legend: '****' ≤0.001 '***' ≤0.005 '**' ≤0.01 '*' ≤0.05 '.' ≤0.1 ' ' ≤1`

Altogether, the recovery status around nine months after the damaging event is worse for households affected by the stronger pathways, i.e. dike breaches in 2013 or flash floods in 2016, compared to the low/medium pathways, i.e. river floods in 2013 or surface water floods in 2016. It should be noted that the financial damage was the most severe for flash floods, while the psychological burden and the perceived recovery were the worst for residents who experienced dike breaches in 2013, who

are then followed by the flash flood cases (Table 3). Maybe the better recovery among severe cases in 2016 is owing to the stronger community resilience that was found to buffer psychological burden in Simbach and surroundings (Masson et al., 2019). Furthermore, it is striking that the average pay-outs for loss compensation are – in relation to the mean financial losses – considerable higher for the cases affected by the 2016-floods in comparison to the 2013-flood.

In general, financial losses, recovery and psychological burden show highly significant differences between the two events as well as between the pathways. Financial impacts and recovery tend to follow the severity pattern of the flood characteristics (i.e. the hydraulic impact variables shown in Table 2), particularly the water level, which is considered the most important variable that explains flood damage (e.g. Gerl et al., 2016; Vogel et al., 2018). Within each flood event, the stronger flood pathway, i.e. dike breaches and strong flash floods, show significantly higher values than their less severe counterparts (river and surface water floods). This supports the hypothesis that the overall (hydraulic) severity of a flood pathway is more important for the perceived psychological burden than the general flood type (see Laudan et al., 2020). The results further support studies that suggest developing pathway-specific loss models (Vogel et al., 2018; Mohor et al., 2020). At this point, the question arises whether and to which degree flood pathways also govern coping options.

**4.4 Short-term response as coping strategy: warning and emergency measures**

There are several strategies to mitigate flood impacts, of which 1) preparedness and response in the case of an event, 2) damage mitigation by implementing property-level adaptation measures and 3) risk transfer in terms of insurance coverage are the most relevant for residents (see Driessen et al., 2016). The first strategy can also be described as reactive or short-term response, while the second is seen as a more proactive or long-term coping strategy (Neise and Revilla Diez, 2019). Insurance coverage does not reduce damage primarily, but facilitates a quick recovery since financial losses are compensated; its interlinkage with property-level adaptation is not clear (e.g. Surminski and Thieken, 2017; Hudson et al., 2017, 2020). In this section, we focus on reactive responses, for which timely warning is an important pre-requisite (e.g. Penning-Rowsell and Green, 2000).

Table 4 reveals highly significant differences between the two flood events with regard to warning and emergency response. Residents affected by the 2013-flood were warned more often and at a considerably longer lead time in comparison to the 2016-event (Table 4). After the extreme flood event in 2002 in Germany, various initiatives and high investments had been undertaken to improve river flood risk management including early warning and preparedness, which had proven successful in 2013 (Thieken et al., 2016a; Kreibich et al., 2017).

Table 5 provides more details on how people had become aware of the imminent flood danger underlining the huge differences between the two flood events. Considerably more residents who had been affected by the 2013-flood received official flood warnings than it was the case in 2016: while 31 to 55% of the people affected in 2013 were warned by severe weather warnings, flood alerts or calls for evacuation, this applies to just 3.4 to 13.7% of those affected in 2016 (Table 5). It is striking that own/independent observations play an important role in all four data subsets: one third to more than half of the people per subsample reported that their own observations of e.g. cloud formations, heavy rainfall or rising water levels





made them aware of the imminent flood danger (Table 5). However, while just 4 to 6% of the 2013-flood victims were not warned at all, this applies to 26 to 35% of interviewed people in 2016 (Table 4 and 5). This reflects the current differences in the warning capabilities of river floods and convective storms or flash floods: while river floods, particularly at reaches

downstream, can be forecasted several days in advance; forecasting convective storms that cause pluvial flooding is more challenging due to the dynamic formation of convective cells. Hence, lead times are restricted to a few hours, if at all (Merz et al., 2020). This is illustrated by the average lead time that is particularly short for the flash floods in 2016 (Table 4). The analyses of the median values suggest that 50% of the people affected in 2013 were warned at least 24 hours before the water entered their home, while this value drops to just one hour for the 2016-subsets (Table 5), which contain a maximum value

of 24 hours. For the pathway dike breach, which is characterized by stronger and unforeseen flooding, the mean lead time is significantly different from the mean value for river floods (Table 4), indicating that dike breaches pose an additional challenge on timely and informative warnings and hence time-critical situations may arise in the hinterland of dikes.

**Table 4: Characteristics of the warning process and emergency response as reported by households affected by different flood pathways in 2013 and 2016.**

| Subsample (Pathway) | River2013 | ↔ | Dike2013 | 2013 ↔ 2016 | Surface2016 | ↔ | Flash2016 | Overall |
|---|---|---|---|---|---|---|---|---|
| Sample size | 1258 | — | 394 | — | 473 | — | 128 | 2253 |
| Households that received no warning [%] | 6.3 | | 4.3 | **** | 34.9 | . | 25.8 | 13.1 |
| Mean warning lead time [hours] | 36.5 | * | 30.4 | **** | 2.5 | | 0.9 | 29.2 |
| Mean perceived knowledge about self-protection [Likert scale from 1 (I knew exactly what to do) to 6 (I did not know at all what to do)] | 2.4 | **** | 3.1 | **** | 4.4 | . | 4.7 | 3.1 |
| Average number of performed emergency measures [count] | 4.24 | *** | 4.64 | **** | 1.67 | | 1.97 | 3.64 |

*Comparison of subsets or 2013 to 2016 in the middle columns, with P-value ranges from Mann-Whitney-Wilcox or Chi-Square tests represented as: Legend: '****' ≤0.001 '***' ≤0.005 '**' ≤0.01 '*' ≤0.05 '.' ≤0.1 ' ' ≤1*

Furthermore, residents affected by river floods in 2013 knew much better how to protect themselves from flooding than

390 people affected in 2016 (Table 4). In addition, the values of the perceived response knowledge indicate highly significant differences within the event of 2013, suggesting that people affected by dike breaches had to cope not only with shorter warning times, but were more often unaware of what they could do to mitigate losses and to protect their lives. Within the 2016-event only slight differences can be detected, but there is an indication that people affected by flash flood were less informed/prepared (Table 4). In detail, the percentage of well-informed people who chose a 1 or 2 when asked how well they

knew how to protect themselves and their household from flood impacts on a scale from 1 to 6, drops from 62.5% for river floods in 2013 to 46.7% in the subset containing dike breaches and even to 22.2% in cases with surface water flooding in

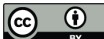

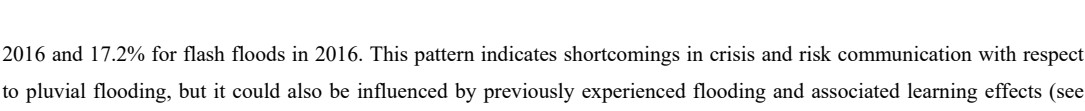

2016 and 17.2% for flash floods in 2016. This pattern indicates shortcomings in crisis and risk communication with respect to pluvial flooding, but it could also be influenced by previously experienced flooding and associated learning effects (see section 4.6).

The different warning capabilities and the different levels of perceived response knowledge are further reflected in the responsive behaviour during the events: residents affected in 2013 undertook a significantly higher number of emergency measures namely around four or five, than those affected in 2016 with one or two measures) on average, while there are no differences between pathways in 2016 (Table 4).

**Table 5: Answers to the question: "How did you become aware of the imminent danger of being flooded?" (multiple answers possible), given in percentage of all interviewed residents per subset and median lead time per subset.**

| Subset | 2013-river flood | 2013-dike breach | 2016-surface water flood | 2016-strong flash flood |
|---|---|---|---|---|
| Severe weather warning (by DWD) | 44.0% | 31.0% | 13.7% | 7.8% |
| Severe weather warning (by other agencies) | --- | --- | 3.4% | 7.0% |
| Flood warning by authorities | 44.7% | 50.0% | --- | --- |
| Warning and evacuation at the same time | 25.5% | 54.8% | --- | --- |
| General media coverage | 16.8% | 18.5% | 4.9% | 2.3% |
| Warning by neighbours, friends etc. | 17.7% | 19.0% | 7.4% | 13.3% |
| Own independent search for information | 27.8% | 24.9% | 1.9% | 0.8% |
| Independent observations (e.g. water levels) | 46.8% | 35.3% | 48.0% | 54.7% |
| No awareness of the imminent hazard (no warning) | 6.3% | 4.3% | 34.9% | 25.8% |
| Not specified / no answer | 1.0% | 0.8% | 2.1% | 1.6% |
| Number of valid cases (warning source) | 1258 | 394 | 473 | 128 |
| Median of the lead time [hours] | 24 | 24 | 1 | 1 |
| Number of valid cases (lead time) | 922 | 305 | 149 | 42 |

To get a clearer picture, Figure 3 shows what kind of emergency measures were undertaken. While residents affected by fluvial flooding in 2013 performed a variety of measures, residents affected in 2016 relied on water pumps. Further, it should

be noted that in the case of fluvial floods electricity and natural gas is more often switched off centrally, while those affected by pluvial floods have to take care for it on their own, which poses further risks of electrocution in case a person enters the water.

Overall, the analyses illustrate that residents in areas that are prone to river flooding were provided with better and timely warning information in 2013. Together with their higher level of response knowledge they were capable of performing more

emergency measures than residents affected in 2016. Since emergency response seems to be an effective coping strategy for pluvial flooding, particularly due to their relative low water depths (Rözer et al., 2016; Spekkers et al., 2017), our analysis highlights that there is room for improving not only early warning, but also communicating potential measures and adequate behaviour in case of pluvial flooding. Table 4 and Figure 3 further reveal that residents affected by dike breaches tend to perform more emergency measures, although they have less knowledge and shorter lead times. The resulting damage (Table



3) shows that this strategy probably only mitigates a small amount of damage. Hence, more studies on the efficacy of emergency measures are needed.

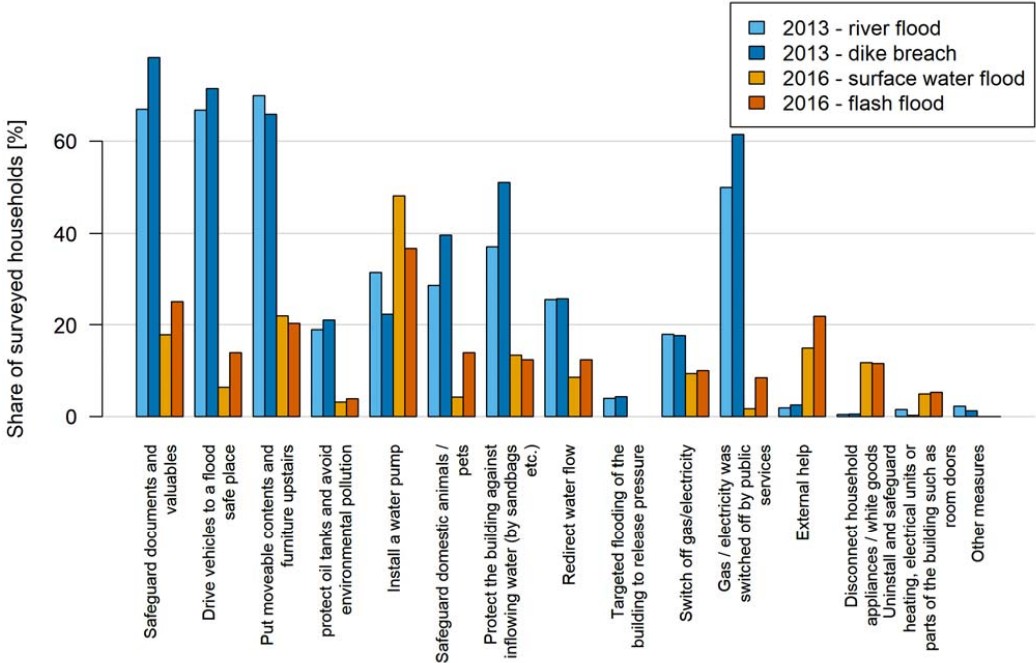

**Figure 3: Performed emergency measures before and during the event as reported by households affected by different flood pathways in 2013 and 2016 (multiple answers possible).**

**4.5 Long-term response as coping strategy: performance of property-level flood adaptation BEFORE and AFTER the floods**

Besides emergency response in the case of an event, there are various proactive precautionary (or adaptive) measures that can reduce flood losses (e.g. Kreibich et al., 2015; Attems et al., 2020a). Both surveys included questions on the actual and intended implementation of property-level flood adaptation. In particular, respondents were asked to state whether they had

implemented a specific measure before or after the event, are planning to do so within the next six months or do not intend to implement that measure. In total, 16 measures were considered, four of which comprised informative measures (search for information about the flood risk or adaptation options, attendance of flood seminars or participation in neighbourhood networks). Another six measures addressed non-structural adaptation (flood-adapted building use, flood-adapted interiors and avoidance of noxious liquids in the cellar, e.g. petrol, paint), which also included measures that improve preparedness

(purchase of a water pump or an emergency power generator or existence of an emergency plan and box). Insurance





coverage was treated separately. Finally, the implementation of five structural measures was studied (i.e. relocating heating and electricity, securing heat and oil tanks, improving the flood safety of the building, installing a backflow preventer or water barriers).

In Table 6 the mean relative implementation per category is presented for the situation before the damaging flood and around nine months later. To calculate the relative implementation, the total count per category was normalized by the count of possible measures per category to gain comparable class results, i.e. a person who has implemented all five structural measures got the value 1, a person who has only secured the heat and oil tank and implemented a backflow preventer received 0.4 (2/5). The values in Table 6 correspond to the average relative implementation of the measures per category per subsample. It should be noted that only property owners were asked about the five structural measures.

Table 6 reveals that adaptive behaviour before the floods was significantly different between the two flood events. In all categories, i.e. informative, non-structural and structural measures, as well as insurance, people affected in 2013 had been better adapted to the flood risk than residents affected in 2016. In most categories, the values for 2013 are around twice as high as in 2016. With regard to the different pathways, there are no differences in the 2016-cases, while in the 2013-samples there's a significant difference with regard to non-structural adaptation and a slight difference in structural adaptation. Hence, people affected by dike breaches in 2013 were less adapted than those affected by river floods (Table 6).

After the flood, the adaptation status and the differences between and within flood events changed considerably revealing a pathway-specific behaviour. Table 6 illustrates a boost of information-seeking behaviour in all subsets with, however, a varying degree: people affected by flash floods in 2016 searched most frequently for additional information on flood risk and mitigation options, followed by people affected by surface water flooding in 2016, dike breaches in 2013 and river floods in 2013 (Table 6).

If we sum up the mean relative implementation of informational precaution before AND after the floods, people affected by dike breaches performed best (52.4% mean implementation), followed by river floods in 2013 and flash floods in 2016 (48.8% mean implementation each) and surface water floods in 2016 (just 35.7% mean implementation). This pattern persists when intended information-seeking behaviour is included (Fig. 4) and illustrates that particularly severe flood pathways and impacts trigger information-seeking behaviour.

When it comes to the implementation of non-structural and structural measures or to the conclusion of an insurance policy, the additional mean implementation follows – in principle – a pattern similar to the information-seeking behaviour: residents affected in 2016 implemented the most additional measures, followed by those affected by dike breaches in 2013 and river floods in 2013 (Table 6). This might also be due to the fact, that more people affected by river floods in 2013 had already implemented measures before the flood, so the perceived necessity for further improvement after the flood was not as high as among residents affected by surface water and flash floods in 2016.





**Table 6: Property-level adaptation before and after damaging floods as reported by households affected by different flood pathways in 2013 and 2016.**

| Subsample (Pathway) | River2013 | ↔ | Dike2013 | 2013 ↔ 2016 | Surface2016 | ↔ | Flash2016 | Overall |
|---|---|---|---|---|---|---|---|---|
| Sample size | 1258 | — | 394 | — | 473 | — | 128 | 2253 |
| *Property-level adaptation (long-term) – BEFORE the flood* | | | | | | | | |
| People who sought information about the flood hazard or protection options [%] | 76.5 | . | 71.8 | **** | 34.4 | | 31.4 | 64.3 |
| Of those without flood experience, people who sought information about the flood hazard or protection [%] | 65.7 | | 70.7 | **** | 26.7 | | 28.1 | 51.8 |
| Mean relative implementation of 4 potential informational precautionary measures [%] | 39.7 | | 37.6 | **** | 13.7 | | 13.5 | 32.4 |
| Mean relative implementation of 6 potential non-structural precautionary measures [%] | 40.7 | *** | 35.4 | **** | 20.7 | | 20.2 | 34.4 |
| Mean relative implementation of 5 potential structural precautionary measures [%] (1) | 20.0 | . | 17.6 | **** | 9.2 | | 9.2 | 16.7 |
| Households that took out insurance [%] | 56.9 | | 56.5 | **** | 36.6 | | 35.2 | 51.4 |
| *Property-level adaptation (long-term) – AFTER the flood* | | | | | | | | |
| Mean relative additional implementation of 4 potential informational precautionary measures [%] | 9.1 | **** | 14.8 | **** | 22.0 | **** | 35.3 | 14.3 |
| Mean relative additional implementation of 6 potential non-structural precautionary measures [%] | 7.6 | **** | 12.1 | **** | 16.5 | | 16.0 | 10.7 |
| Mean relative additional implementation of 5 potential structural precautionary measures [%] (1) | 5.6 | ** | 7.9 | **** | 15.9 | | 16.3 | 8.8 |
| Households that took out insurance after the flood [%] | 4.7 | *** | 9.1 | **** | 13.5 | **** | 26.4 | 8.6 |

*Percentage or means only regarding valid values, i.e. answered entries.*

*(1) Only among homeowners*

*Comparison of subsets or 2013 to 2016 in the middle columns, with P-value ranges from Mann-Whitney-Wilcox or Chi-Square tests represented as: Legend: '****' ≤0.001 '***' ≤0.005 '**' ≤0.01 '*' ≤0.05 '.' ≤0.1 ' ' ≤1*



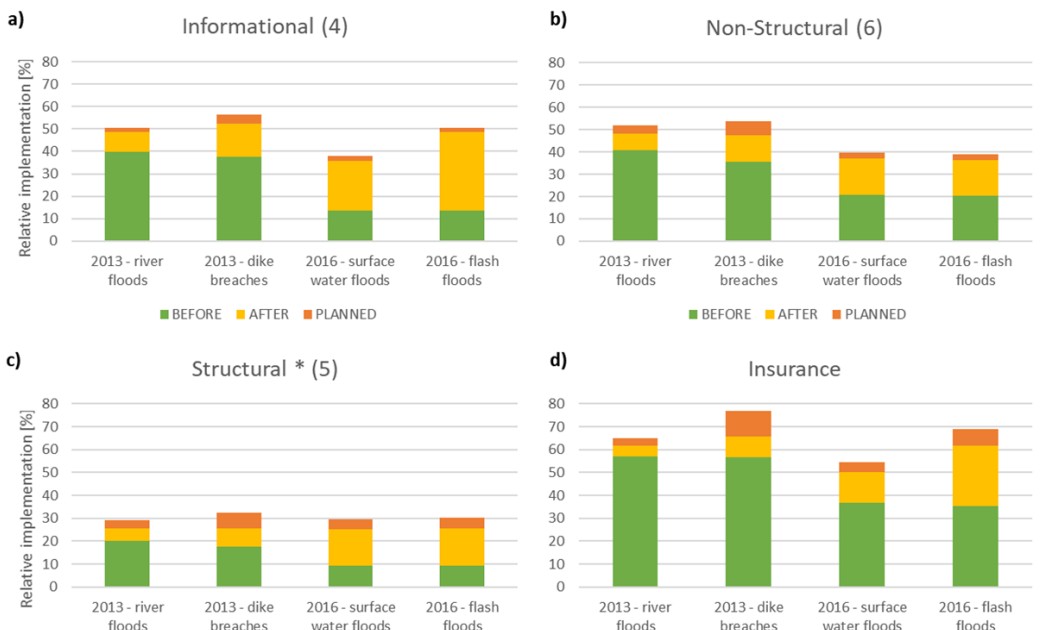

**Figure 4: Cumulative mean relative implementation of adaptation, including measures that were (at the time of the survey) planned to be implemented within the next six months (\* surveyed only among homeowners).**

Considering that the subgroups started at very different levels of adaptation before the events stroke, the cumulative implementation depicted in Fig. 4 reveals that non-structural measures are more popular along rivers, i.e. among those affected in 2013. On average, a relative implementation of 50% of six measures is reached, meaning that on average three measures have been implemented per affected household, in contrast to around 40% or 2.4 measures in the case of the 2016-subsamples (Fig. 4b). Maybe this is due to a higher risk perception of fluvial floods in contrast to pluvial floods.

Interestingly, the cumulative implementation of structural adaptation measures reaches a similar level across all four flood pathways, though the overall lowest numbers in comparison to the other categories: around nine months after the floods a mean implementation of around 25% (or 1.3 measures) is reported in all four subsamples and inches up to 30% (or 1.5 measures) when intended adaptation is included (Fig. 4c). This pattern was described before for fluvial floods (Thieken et al., 2007; Kienzler et al., 2015) and pluvial floods (Rözer et al., 2016), were structural measures such as sealing the

basement, relocating heating or electrical utilities to higher stories or changing the heating system or protecting the oil tank had been identified as the least popular measures. Most likely this is due to the higher costs of structural measures.



The conclusion of insurance reflects the pattern of the information-seeking behaviour (Table 6, Fig. 4d) and highlights that particularly people who experienced severe flood pathways strive for a backstop. In addition, the severity of the flood processes and their impacts might cause a lower appraisal of the efficacy of adaptive measures on the property-level. Therefore, the next section finally looks at perceptions.

**4.6 Previously experienced flooding and risk perceptions**

Since previously experienced floods impact risk perceptions and influence adaptive behaviour (e.g. Bubeck et al., 2012), Table 7 summarizes data on previously experienced flooding as well as average perceptions of flood risk, coping capacities and responsibilities for flood risk reduction. There are significant differences with regard to previously experienced flooding between and within the events (Table 7). Most households affected by river floods in 2013, i.e. 64%, had been previously affected. This percentage is much lower in the subset on dike breaches in 2013 (34%) as well as on surface water floods in 2016 (29%), and flash floods (only 19%; Table 7). As noted by Kienzler et al. (2015), having experienced river floods has considerably changed after the 2002-flood, whilst there is a lower percentage among those affected by pluvial floods, which was also observed by Spekkers et al. (2017), who reported that just 21% of households surveyed in the city of Münster had been flooded before the severe pluvial flood of July 2014.

However, among all surveyed households, less than 15% had experienced a flood in the ten years preceding the events that are studied in this paper, with a distinction between the stronger pathways (8% and 5% for 2013-dike breaches and 2016-flash floods subsamples, respectively) and the low/medium flood pathways (16% for 2013-river and 17% for 2016-surface water floods subsamples, see Table 7). Residents that were affected by flash floods in 2016 were the least experienced with flooding: just 19% had been flooded before and only 5% over the ten years preceding that flood. It is remarkable that the highly significant differences in previously experienced floods between the two events vanish when just the preceding ten years were taken into account, while the differences within the events, i.e. between the different pathways, remain (Table 7). With regard to various perceptions, it is striking that there are no differences between the events nor between the pathways with regard to perceived self-efficacy and just small differences with regard to the perceived responsibility of the government (Table 7). A comparison with other regions and data could reveal whether the reported values could be regarded as representative mean perception. Particularly, self-efficacy is seen as a key component for adaptive behaviour (Bubeck et al., 2013; Poussin et al., 2014).

Average perceived response costs, response efficacy and responsibility of any individual to reduce damage, however, differ between the two events: people affected in 2013 perceived response costs a bit higher than those affected in 2016; this also holds for the perceived efficacy of measures and the responsibility of individuals (Table 7). It is striking that response efficacy is perceived the lowest by people who were affected by flash floods in 2016, probably highlighting the high velocities and severe impacts on buildings and indicating the limits of property-level adaptation (see above).



**Table 7: Previously experienced flooding and perceptions on risk and property-level adaptation of households affected by different flood pathways in 2013 and 2016.**

| Subsample (Pathway) | River 2013 | ↔ | Dike 2013 | 2013 ↔ 2016 | Surface 2016 | ↔ | Flash 2016 | Overall |
|---|---|---|---|---|---|---|---|---|
| Sample size | 1258 | — | 394 | — | 473 | — | 128 | 2253 |
| **Previously experienced flooding** | | | | | | | | |
| People who experienced at least one previous flood [%] | 63.9 | **** | 33.7 | **** | 29.3 | * | 19.0 | 48.6 |
| People who experienced a flood in the ten years preceding the damaging event [%] | 16.4 | **** | 7.5 | | 17.0 | **** | 4.8 | 14.3 |
| **Perceptions – AFTER the flood** | | | | | | | | |
| Mean perceived probability of future floods [Likert-scale from 1 (it is very UNlikely that I will be affected by future floods) to 6 (it is very likely that I will be affected by future floods)] | 4.6 | **** | 3.8 | **** | 3.7 | **** | 3.1 | 4.2 |
| Mean perception of impacts of a future flood LIKE THIS ONE [agreement to the statement "It won't be as bad as in 2013/16" 1: I fully agree to 6: I fully disagree] | 4.1 | **** | 3.6 | . | 4.1 | **** | 3.1 | 3.9 |
| Mean perceived self-efficacy [agreement to the statement "Personally, I am UNable to implement any of the proposed precautionary measures" 1: I fully agree to 6: I fully disagree] | 4.3 | | 4.2 | | 4.5 | | 4.2 | 4.3 |
| Mean perceived costs [agreement to the statement "Private precautionary measures are too expensive" 1: I fully agree to 6: I fully disagree] | 2.9 | | 2.9 | *** | 3.3 | | 3.0 | 3.0 |
| Mean perceived response efficacy [agreement to the statement "Private precautionary measures can considerably reduce damage" 1: I fully agree to 6: I fully disagree] | 2.6 | . | 2.8 | *** | 2.7 | **** | 3.4 | 2.7 |
| Mean perceived responsibility of the government [agreement to the statement "Flood risk reduction is a task of the government, not of the residents" 1: I fully agree to 6: I fully disagree] | 3.0 | * | 2.8 | . | 3.1 | | 2.9 | 3.0 |
| Mean perceived responsibility of individuals [agreement to the statement "Everyone is obliged to reduce flood damage as much as possible" 1: I fully agree to 6: I fully disagree] | 1.7 | * | 1.9 | **** | 2.4 | | 2.6 | 1.9 |

*Percentage or means only regarding valid values, i.e. answered entries.*

*(1) Only among the homeowners*

*Comparison of subsets or 2013 to 2016 in the middle columns, with P-value ranges from Mann-Whitney-Wilcox or Chi-Square tests represented as: Legend: '****' ≤0.001 '***' ≤0.005 '**' ≤0.01 '*' ≤0.05 '.' ≤0.1 ' ' ≤1*

Furthermore, the threat appraisal of future floods differs significantly between and within the events: the perceived probability of future floods is the highest among residents affected by river floods in 2013, followed by dike breaches in 2013 and surface water flooding in 2016. Those who were affected by flash floods in 2016 tend to believe that they will not

be affected again. This pattern is even more pronounced when a statement on the perception of the impacts of a future flood LIKE THIS ONE was assessed: here the ones who were damaged the most (i.e. by flash floods in 2016 and by dike breaches in 2013, see Table 3) tend to think that impacts comparable to the just experienced are less likely to occur (Table 7). This highlights that it is important to distinguish probability and impacts in threat appraisals as shown by Bubeck et al. (2013). The statement on the perceived impacts of future floods also contains a nuance of denial of the flood risk, which might

explain the lower adaptation (see Fig. 4).

## 5 Conclusions

Based on two surveys among residents in Germany who were affected by flooding in 2013 and 2016, respectively, this paper looked at differences in flood processes, impacts and coping strategies between four flood pathways found in these spatially compound flood events. While the socio-economic characteristics did not differ much between the samples (except for

income, which can be explained by the spatial patterns of the floods), impacts and coping strategies differed considerably. Based on the detailed analyses of a broad range of variables, each flood pathway can be characterized qualitatively as shown in Figure 5.

| Characteristics | Flood pathways | | | |
|---|---|---|---|---|
| | river flood 2013 | dike breach 2013 | surface water flood 2016 | flash flood 2016 |
| Hydraulic flood characteristics | MEDIUM | HIGH | SMALL | HIGH |
| Financial impacts | MEDIUM | HIGH | SMALL | HIGH |
| Perceived recovery | MEDIUM | BAD | GOOD | BAD |
| Warning and emergency response | GOOD | MEDIUM | BAD | BAD |
| Property-level adaptation before | GOOD | GOOD | BAD | BAD |
| Property-level adaptation after | MEDIUM | GOOD | BAD | MEDIUM |
| Flood experience | HIGH | MEDIUM | MEDIUM | LOW |
| Risk and reponsibility perception | MEDIUM | MEDIUM | MEDIUM | LOW |

**Figure 5: Qualitative summary of the flood pathway characteristics, where medium often reflects to the averages.**





**River floods (2013)**: The flood processes are characterized by high water levels and long durations of inundations. The financial impacts, recovery and the psychological burden from the flood represent more or less the average of the total data (note that this was the biggest subsample). Most of the residents affected by river floods in 2013 were warned in advance with comparatively long lead times. They were well-prepared, i.e. performed many emergency measures and also showed the highest level of flood adaptation at their property before the flood. After the flood they undertook considerable additional

adaptation, but Fig. 4 shows that they lost their top position and other subsamples reached the same level, although this group believes on average to be affected by future floods and also agrees that individuals have to contribute to flood risk reduction. Overall, adaptation of this group could be supported by financial incentives and funds since they perceive response costs as rather high (Table 7). Such costs might also be related to the efforts involved to implement a measure. Therefore, improved consultation and support during implementation as also proposed by Attems et al. (2020b) deserve

further attention. Since previously experienced flooding was the highest in this subsample, their level of adaptation after the flood might also indicate a kind of saturation level. This hypothesis, however, needs to be researched in more detail.

**Dike breaches (2013)**: This pathway is characterized by very high water levels, very long durations of inundations and a high share of oil contaminations. Consequently, the financial impacts are the second highest, repair works at buildings are slow and the psychological burden from the flood is the highest across all four sub-samples. Most of the residents affected

by dike breaches in 2013 were warned in advance with comparatively long lead times. Like those that were affected by riverine flooding, they performed many emergency measures and showed to be comparatively well-informed about flood hazards and coping options. With regard to structural and non-structural measures, adaptation before the flood was lower than in the river-flood-sample, but they reached a similar level after the flood and a higher level of insurance penetration. Perceptions of flood risk, coping options and responsibilities represent more or less an average behaviour. The fact that

losses are very high despite a good responsive and adaptive behaviour indicates the limits of individual adaptation in view of the high hydraulic impacts caused by dike breaches. Insurance serves as a backstop. Overall, this group should be further educated with regard to risks and suitable coping options. Since response time might be limited in case of dike breaches, potential environmental risks due to bursting oil tanks or the release of other harmful substances should receive particular attention. During the last revision of the German Federal Water Act a regulation of oil tanks in (potentially) flood-prone

areas was already introduce. Still, more information on effective and suitable property-level adaptation is needed.

**Surface water floods (2016)**: The flood processes are characterized by (very) low water levels and short durations of inundations. Financial impacts and psychological burden from this pathway were the lowest across the sub-samples, while there was a speedy recovery. Threats may occur from high velocities. Since most of the residents affected by surface water flooding in 2016 were not warned in advance, lead times were short and knowledge about self-protection was below-

average, people prone to pluvial flooding – this is in general the urban population, since pluvial floods are ubiquitous – should be better informed about potential traps (cellar, subways, cars etc.) and suitable adaptation measures, particularly after events. In comparison to other sub-samples, this group was the least informed and the least insured. Moreover, implementation of non-structural measures was below average – also after the event. Therefore, risk communication has in





general to be improved and has a good chance to be successful since threat and coping appraisals are well developed and the
uptake of measures after the 2016-event was good. Responsibility and feasibility should be clearly communicated and
demonstrated by best practise examples. Workshops could serve as a good instrument in this case as they strengthen self-
efficacy and protection motivation (Heidenreich et al., 2020).

**Flash floods (2016)**: The flood processes are characterized by (very) high water levels and often (very) high flow velocities
which might be accompanied by contamination. These dynamic processes led to the highest financial impacts and a high
psychological burden. Recovery was comparable to the dike-breach-sample of 2013, although this group received the highest
financial support. Like other affected residents in 2016, most of the people in this group were not warned in advance, and if
so, lead times were short. Their preparedness and adaptation before the flood is comparable to the pluvial-flood-2016-group.
After the flood the information-seeking behaviour was very high, as was the conclusion of suitable insurance policies that
serves as a backstop. To strengthen property-level adaptation, risk communication should focus on the efficacy of measures
that can also withstand high flow velocities.

Altogether, the study demonstrates that flood impacts and coping option differ between and also within flood events. The
concept of spatially compound events is helpful to understand different impacts, but also coping and adaptive behaviour.
This leads to the above-mentioned flood pathway-specific recommendation for risk communication and management. The
efficacy of emergency and precautionary measures with regard to different pathways needs further research. Our results
indicate that residents affected by strong pathways such as dike breaches and flash flood perceive limits of their adaptation
options.

**Author contributions**

AT, HK, and MM developed the questionnaires and conducted the surveys. AT and GM designed the statistical test protocol
and GM mainly carried them out. AT mainly prepared the manuscript with contributions from all co-authors. All authors
commented on previous versions of the manuscript. All authors read and approved the final manuscript.

**Competing interests**

The authors declare that they have no conflict of interest.

**Acknowledgements**

The two surveys were undertaken within the BMBF-project "Hochwasser 2013" (FKZ: 13N13017) and the DFG Research
Training Group "NatRiskChange" (GRK 2043/1), respectively. Funding is gratefully acknowledged, as are additional funds
provided by the GeoForschungsZentrum Potsdam and the Deutsche Rückversicherung AG, Düsseldorf, for the data





collection. Both surveys were conducted by explorare GmbH, Bielefeld, Germany. We further thank Sarah Kienzler, Jonas Laudan, and Ina Pech for their contributions to the survey design and data processing. The second author would like to thank the German Academic Exchange Service (DAAD) for a scholarship (Graduate School Scholarship Programme, 2017-ID
615    57320205).

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
