# Peer review of "Compound inland flood events: different pathways—different impacts—different coping options?"

_Natural Hazards and Earth System Sciences, 2021_

## Author Comment (AC1)

**Manuscript "Compound flood events: different pathways–different impacts–different coping options?" by Annegret H. Thieken et al., Nat. Hazards Earth Syst. Sci. Discuss., 2021**

**Comments of Referee 1**

*Manuscript presents an analysis of the database on questionnaires after the 2013 and 2016 floods in Germany, comprehending data on socio-demographic characteristics, hydraulic flood characteristics, financial impacts and recovery, coping strategies, previous experience and risk perception by citizens. The analysis is based on averages of the data within four distinct hazard classifications (=pathways: dike breach, river flood, surface water flood, flash flood); cross-comparisons are discussed on a qualitative basis; no quantitative correlations are performed.*

**Answer 1**: Thank you for this summary of our work, which is in principle fine. However, we have a different view on what is qualitative and quantitative research. We associate qualitative research with research that deals with in-depth exploration (of written, visual or oral material such as interviews), interpretation and narrative and often involves only a small number of cases. In our paper, the cross-comparisons between the four pathways are based on quantitative data with 2253 cases located across Germany. With our analyses we explore the hypothesis that not only different flood types but also different pathways lead to different flood impacts and that therefore also different coping and risk management strategies are necessary, which need to be better considered in risk communication. Differences between flood pathways were tested for significance as we stated on p. 8, line 223–225: "Data subsets were compared either through the nonparametric Mann-Whitney-Wilcoxon two-sample test or Chi-Squared contingency table test, depending on whether a variable was metric or categorical (Noether, 1991), comparing the median of differences or the closeness of expected frequencies, respectively." Therefore, we consider our study as quantitative research. The approach and hypothesis will be outlined clearer in the revised paper.

Further, we don't see how or where correlation analyses could provide further insights, but we are happy to consider your ideas if specified more explicitly. What kind of correlations (between which variables) were you thinking of?

*Paper is in general well organized and written. Some sections may be shortened, but this is not a crucial point. Resulting evidences are sometimes quite expected (for example: warning is more critical for flash flood with respect to river floods; forecasting convecting storms or dike breaches is challenging). Other results are less obvious and potentially more informative.*

*Data analysis reveals some interesting points. However, I think that the global result is weak, and the take-home message not really sharp. I try to elaborate this. The four scenarios (pathways) here depicted have, indeed, significant differences for some of the analysed characteristics. However: are such different a general property of the pathway, or are they related to the specific event? In other words: can we conclude, say, that all (most) dike-breach events have some characteristics differentiating them from all (most) surface water floods?*

**Answer 2**: As you said, some results, e.g. on warning, could be expected. Still, our analysis does not only confirm some of our expectations on this issue, but also delivers quantitative information on the differences between the flood pathways. With regard to warning, we see, for example, that only 4–6% of people affected by fluvial flooding in 2013 were not warned,

while this share rises to 26–35% in the pluvial flood of 2016, which underlines the dimension of the problem when we look at convective storms. Of course, these numbers are only valid for the events studied, but could be used as metric (or benchmark) when comparing different warning performances. We will add this along with some examples in the revised paper.

In our view, empirically based event studies cannot answer the research question at hand universally, but they add to the knowledge about the topic studied and are certainly valuable next to modelling studies, e.g. for a better risk management on the ground or for providing information on reasonable model parameters. In addition, we would like to emphasize that our data set is comparatively huge and covers a relatively large and diverse geographical area. We expect our set of 2253 data points, which were gathered in 249 different municipalities spread across almost all federal states of Germany (i.e. 14 of 16), to be generalizable enough to deliver insights on the characteristics of flood pathways, their impacts and tendencies of coping and adaptive behaviour. We will add more detailed information on the geographic distribution of our data in the revised manuscript.

*As said, some characteristics are quite obviously related to the definition of the pathway, and can be expected to be general (but this would be a relatively expected conclusion); others may not be obvious, but I do not know how to conclude that they have general validity. For example: "shortcomings in crisis and risk communication" are related to pluvial flooding (as suggested at line 398) or to the specific event and the local organization of the specific area?*

**Answer 3**: In addition to answer 2, we would like to emphasize that the findings and statements in all the analyses of this manuscript are not based on the analysis of an isolated locality, which would just reflect very specific local conditions and actions of one local organisation, but on German-wide datasets. Each finding for one pathway is compared to another pathway and is put into a context (of the flood event). Since flood management has in principle been in place in all of Germany for decades, and is mainly the responsibility of the individual federal states or even of cities and municipalities in the case of pluvial flooding, it can be assumed that shortcomings in crisis and risk communication with regard to pluvial flooding are not just related to the event of 2016 or just one local organization. As shown in the manuscript, crisis and risk communications (or: warning, risk awareness and preparedness) differ between different flood pathways across several federal states. If in another region the overall risk communication and management is perhaps higher than in the regions covered by our data, we would still expect that the risk communication towards pluvial flooding would be less thorough in comparison to fluvial flooding, since pluvial flooding is still a new (policy) issue in many cities.

*Aim of the paper is to "reveal whether and how people affected by different flood types and pathways were prepared before the damaging event, how they were impacted in hydraulic, financial and psychological terms, and how they coped with and recovered from these impacts. The intention is to provide more insights that help establish risk management strategies tailored to different flood types and pathways."*

*As we see, the aim is basically to qualitatively describe some general characteristics; as said, I fear that such scenarios may have no general value and, therefore, may not be applicable in similar hazard pathways but in different social and geographic contexts.*

**Answer 4**: As mentioned above (see answer 1), we provide quantitative results about the differences and commonalities in impacts and coping options between flood types and

pathways. We used a large data set covering basically the whole of Germany, statistical analyses, and hypothesis testing.

In addition, we would like to highlight – and we will strengthen this point in the revised manuscript – that our data cover a huge range of social and geographic contexts in Germany as our two surveys contain cases from 14 (out of 16) federal states (Hamburg and Berlin are missing; see Figure 1). While some states are only represented by a few cases, the federal states with a huge number of cases represent different regions, landscapes and socio-economic conditions as well as different organisations responsible for flood risk management in Germany. We will add a few sentences and numbers on this issue in the revised paper.

*Consequently, we may surely agree on the risk management strategies proposed for the four depicted situations, but we should link them to the global scenarios for the four situations, including all hazard + exposure + vulnerability indicators, rather than to the hazard pathways only.*

**Answer 5**: We are not sure whether we fully understood this comment and the meaning of global scenarios. As risk management has to be implemented and executed locally, we don't agree that this issue necessarily needs to be linked to global scenarios. In our view, global studies tend to be broad and maybe too generic: studies that include all regions AND indicators on hazard, exposure and vulnerability end up, necessarily, with simplifying indicators and aggregating regions, losing much of the details that we provide in our manuscript.

Further, we would highlight again that our data cover a broad range of exposure and vulnerability indicators, but the analysis of the hydraulic impacts shows that the hazard characteristics between the pathways are very different (see Table 2), while the socio-demographic characteristics are not (Table 1). Still, the (financial) impacts mainly reflect the patterns of the hydraulic characteristics (Table 3). We may conclude from this finding that hazard pathways need to be better considered in risk management and communication and are maybe more important than exposure and vulnerability characteristics. However, this has to be analysed in more detail.

*Finally, I have one specific (but not irrelevant) objection: definitions of the pathways are not univocal. Distinction between surface water flood and flash flood for the 2016 event, even if somehow subjective, looks relatively robust. However, distinction between dike breach and river flood for the 2013 event is based on the indication of citizens: "all households that reported that they had been affected by a dike breach were included in this subsample". Can we really assume that citizens have robust knowledge of the mechanics of the flood (pathway) hitting their house? Was any validation of their declarations performed?*

**Answer 6**: We agree that survey answers are associated with uncertainties. To reduce them we conducted both surveys 8 to 10 months after each event under study, which allowed the residents to not only properly assess their losses and to recover from them, but also to better understand the event that happened. In addition, large samples help to obtain robust results. We have close to 400 dike breach cases (i.e. 394). Even if a few of them have wrongly related the inundation at their property to a dike breach, we expect the analyses results to be robust: we crosschecked the answers with the known locations of severe dike breaches in 2013, which are mentioned in the paper in the lines 113-118: (1) a breach at Deggendorf-Fischerdorf, (2) a breach in Klein Rosenburg-Breitenhagen and (3) a breach near Fischbeck. Based on zipcodes and place names, at least 74, 129 and 62 cases that reported dike

breaches in our datasets can be clearly linked to these three breaches, respectively. We will add this information in the revised paper.

*On the whole, I do not think that the manuscript provides significant innovation and extra knowledge for the field of flood risk assessment and management.*

**Answer 7**: What a pity that you don't see the value of our data and our analyses. We are convinced that event analyses are essential for natural hazards research, since there are no other means of gaining more knowledge about risk processes and management options. Even pure modelling studies rely on empirical data from events for model calibration and validation.

We believe that the following two aspects make event analyses particularly valuable and enable researchers to gain new knowledge (e.g. in contrast to a limited case study analysis): 1) reasonably large datasets; 2) theoretical grounding and hypothesis testing. Both aspects are given in our study: 1) our dataset comprises 2253 damage cases from two large scale flood events in Germany, covering 249 municipalities located in 14 (of 16) federal states. As such, different regions, landscapes and socio-economic conditions are covered where different organisations are responsible for flood risk management in Germany, since this is the responsibility of the states (or even of the cities and municipalities in the case of pluvial flooding). 2) Our analyses are based on the hypothesis, that not only different flood types but also different pathways lead to different flood impacts and that therefore also different coping and risk management strategies are necessary, which need to be better considered in risk communication. Your review helped us to sharpen these aspects. We hope that you appreciate our responses.

---

## Author Comment (AC2)

**Manuscript "Compound flood events: different pathways–different impacts–different coping options?" by Annegret H. Thieken et al., Nat. Hazards Earth Syst. Sci. Discuss., 2021**

**Comments of Referee 2**

*This is a simple but useful article that is using historical flood loss data to compare flood types. It presents observations such as that dike breaches typically have longer flood durations or that flash floods typically have higher flow velocities. Many of the conclusions are obvious but its good to have data that confirms these intuitions. I'm not aware of another study doing the same and I think no other dataset would be more suitable for such a study. The article is very well written but I do have some minor comments.*

**Answer 1**: Thank you for this positive assessment of our manuscript.

*Are the samples independent enough to draw generalizable conclusions? Could some of the conclusions not be statistical significant because of a high spatial correlation between the samples? For example, you only have 128 flash flood samples but over how many different locations (e.g. villages) are they really collected? Some of the variables you look into could be highly spatial correlated (e.g. the same for the whole village or even the entire flood event). The most extreme example is table 5 which is somewhat correlated throughout the event (e.g. media coverage, quality weather forecast, etc.). So arguable the sample size for table 5 is just 1? I think this drawback of the study should be highlighted more throughout the results and discussion section so that the readers know which conclusions can be generalized to other areas/countries and which conclusions could be just a local coincidence.*

**Answer 2**: Fig. 1 in the paper shows the wide geographic distribution of our data. In total, households from 249 municipalities located across 14 out of the 16 federal states of Germany have answered the survey. We now had a closer look and found that the smallest subset of 128 cases (flash floods) comes from eight different municipalities, distributed across four different federal states (see again Figure 1). We will add this information in the revised manuscript.

The geographic distribution also holds for the analysis shown in Table 5. For example, the information about the lead time in the flash flood subset was retrieved from 42 cases in six different municipalities located in three different federal states (Bavaria, Baden-Wurttemberg and Rhineland-Palatinate). Early warning and flood risk management in Germany is in the responsibility and thus separately organized by the federal states. And the early warning chains might differ between municipalities.

*The definition of flash flood is a bit subjective in this paper and the conclusions will be very sensitive for this classification. It seems like some circular reasoning could be occurring. That is you seemed to have used some flood intensity information from the household surveys to label observations a flash flood and then you seem to have concluded that flash floods are typically more intense in the same survey data. Is this observation correct or did you merely validate your flash flood classification with the household survey? Could you discuss the potential consequences of your labeling technique on conclusions of the paper? Furthermore, I wonder whether there isn't a more objective way of classifying flash floods. Maybe extreme rainfall in a terrain that isn't flat? Have you done some literature review on this?*

**Answer 3**: As described in section 3 (line 209-221), we used EXTERNAL data to distinguish pluvial from flash floods. In fact, we started with rainfall data from the German Meteorological Service (DWD) for a first classification, in which we selected places that experienced rainfall exceeding the severe weather warning threshold of hourly rainfall (25 mm/h) – a method that has been applied in other studies, too, e.g. in a DWD-GDV project on heavy rainfall (Winterrath et al. 2017). We continued to verify the rainfall-driven selection of affected places by media reports. In a last stage, we used the survey data for a validity check of our classification. In fact, no changes were made after this cross-check. We will double check our classification by including terrain/topographic information and will rephrase the section (lines 209-221) accordingly.

The following reference will be added to the paper:

Winterrath, T., Brendel, C., Hafer, M., Junghänel, T., Klameth, A., Walawender, E., Weigl, E. & Becker, A. (2017): Erstellung einer radargestützten Niederschlagsklimatologie, Berichte des Deutschen Wetterdienstes Nr. 251, Deutscher Wetterdienst, Offenbach am Main. https://www.dwd.de/DE/leistungen/pbfb_verlag_berichte/pdf_einzelbaende/251_pdf.pdf?__blob=publicationFile&v=2

Finally, we would like to highlight that the hydraulic characteristics shown in Table 2 confirm the expected differences between the pathways, although the survey data on these hydraulic characteristics were NOT used to classify the cases. So, there is no circular reasoning in our view.

*The use of the term "compound flood event" is causing confusion. From just reading the title most readers would assume this is about the coincidence of coastal and fluvial flooding. This makes the paper title somewhat misleading and also confusing because the link to the rest of the title is then no longer clear. The abstract adds to this confusion as the term compound flooding is introduced in an unexpected context. The start of section 2 clarifies everything very well and I understand why the term is appropriate but I still recommend either explaining the unconventional use of the term compound flood event early in the abstract or using different terminology (e.g. why not use the word flood type in the title).*

**Answer 4**: Thank you for this suggestion. In fact, we changed this several times in earlier versions of the manuscript and obviously introduced some inconsistency here. We propose to change the title of the paper to "Compound **inland** flood events: different …" and to explain the term "Compound **inland** flood events" in the abstract to avoid confusion.

*Section 3 doesn't explain the approach at a high level. This approach is quite simple so you can keep it short.*

**Answer 5**: We will further shorten section 3 in the revised paper.

---

## Author Response (AR2)

**09 Nov 2021**

**Editor decision: Publish subject to minor revisions (review by editor)**

**by Bart van den Hurk**

Comments to the author:

The two reviewers have a different recommendation on how to proceed with the paper. Both reviewers indicate that the revised manuscript contains a lot of interesting evidence that will inspire future research scholars in how to address various sources of diversity in the analysis of flood damage and consequences. However, reviewer #1 also notes that the conclusions from this analysis are fairly generic and don't reveal a clear picture on how these results should be taken forward in analysis of flood impacts, particularly in (cultural) settings outside Germany. Although it is not within the scope of this manuscript to describe a worldwide assessment of flood impacts, it would be very valuable if the conclusion section would contain a set of statements on how the results reported here could not only refer to the complex interplay of factors as illustrated, but also could help to make this complexity better manageable while designing early warning, response or educational applications, preferably taking into account the unknown factors contributing to complexity in areas (outside Germany) where this survey was not carried out. This new version will not be sent out for external review anymore.

**Response as of 18 Nov 2021**

Dear editor,

Thank you for considering our paper for publication in NHESS. We carefully read the manuscript again, changes a few minor things in the introduction and following sections and focussed on providing more substantial conclusions. The last part of the concluding section was substantially revised and extended. It now reads:

*"Altogether, the study demonstrates that flood hazard characteristics, impacts and coping options differ between and also within compound inland flood events. Hydraulic characteristics and flood impacts are strongly governed by the specific flood pathway, while coping options (short and long term) are more related to the general flood type (i.e., fluvial and pluvial). Hence, the concept of spatially compound events is helpful to understand different flood impacts, but could be strengthened towards coping and adaptive behaviour. The above-mentioned flood pathway-specific recommendations for risk communication and management are a first step in this direction. In addition, we can draw some conclusions that go beyond the studied cases and the German context.*

*First, the relation between hydraulic forces and impacts strongly support recommendations of developing pathway-specific loss models as done by Vogel et al. (2018) or Mohor et al. (2020, 2021). Research on this is, however, in its infancy. Secondly, to further mitigate damage, risk and crisis communication should distinguish not only flood types, but also pathways highlighting their specific threats, e.g. life-threatening situations during flash floods. Identifying and communicating such threats might better fulfil user needs, as it has been shown that adding impact information or additional descriptions of the threats may provide a clearer picture of the upcoming situation than abstract indications of warning levels (e.g. "strong"), specially to less proficient users (Kox et al., 2018). With regard to flash floods options for local warning and alerting systems should be explored as an option of improving warning and response in small catchments.*

*Thirdly, it should be noted that experiencing strong flooding caused by dike breaches or flash floods boost precaution, while surface water flooding does not, although the latter can happen almost everywhere. Therefore, modes to communicate and experience flood impacts in a tangible way are particularly important (e.g. exhibitions, storytelling etc.). In addition, the efficacy of emergency and precautionary measures with regard to different pathways needs further research. Finally, people affected by strong pathways such as dike breaches or flash floods (with sediment loads) need special assistance to recover physically and mentally from the impacts; their burden is the highest. Our results indicate that these residents experience limits of their adaptation options."*

We hope that these amendments meet your expectations.

Thank you for your patience with this paper.

Best regards

Annegret Thieken

(on behalf of all authors)